# Limited Impact of Delta Variant’s Mutations on the Effectiveness of Neutralization Conferred by Natural Infection or COVID-19 Vaccines in a Latino Population

**DOI:** 10.3390/v13122405

**Published:** 2021-11-30

**Authors:** Carlos A. Sariol, Crisanta Serrano-Collazo, Edwin J. Ortiz, Petraleigh Pantoja, Lorna Cruz, Teresa Arana, Dianne Atehortua, Christina Pabon-Carrero, Ana M. Espino

**Affiliations:** 1Department of Microbiology and Medical Zoology, University of Puerto Rico-Medical Sciences Campus, San Juan, PR 00936, USA; lorna.cruz@upr.edu (L.C.); teresa.arana@upr.edu (T.A.); ana.espino1@upr.edu (A.M.E.); 2Unit of Comparative Medicine, University of Puerto Rico-Medical Sciences Campus, San Juan, PR 00936, USA; crisanta.serrano@upr.edu (C.S.-C.); edwin.ortiz11@upr.edu (E.J.O.); petraleigh.pantoja@upr.edu (P.P.); 3Department of Internal Medicine, University of Puerto Rico-Medical Sciences Campus, San Juan, PR 00936, USA; 4Puerto Rico Science, Technology and Research Trust, San Juan, PR 00927, USA; dianne.atehortua@prcci.org (D.A.); cpabon@prcci.org (C.P.-C.)

**Keywords:** SARS-CoV-2, COVID-19 vaccine, neutralization, serology, protection

## Abstract

The SARS-CoV-2 pandemic has impacted public health systems all over the world. The Delta variant seems to possess enhanced transmissibility, but no clear evidence suggests it has increased virulence. Our data show that pre-exposed individuals had similar neutralizing activity against the authentic COVID-19 strain and the Delta and Epsilon variants. After only one vaccine dose, the neutralization capacity expanded to all tested variants in pre-exposed individuals. Healthy vaccinated individuals showed a limited breadth of neutralization. One vaccine dose did induce similar neutralizing antibodies against the Delta as against the authentic strain. However, even after two doses, this capacity only expanded to the Epsilon variant.

## 1. Introduction

Coronavirus disease 2019 (COVID-19), caused by the novel severe acute respiratory syndrome coronavirus 2 (SARS-CoV-2), is responsible for the most recent global pandemic, declared by the World Health Organization (WHO) on 11 March 2020 [1]. COVID-19 is a serious illness that has had a significant impact on public health due to its high morbidity and mortality rates [2]. At the beginning of the pandemic, most countries around the world underwent a strict lockdown that had serious social, economic and health service effects. As a consequence, many non-COVID-19 diseases, such as tuberculosis [3] and cancer [4], suffered delays in diagnosis and treatment, leading to more severe illness and outcomes.

According to a WHO report, as of 22 November 2021, there had been 256,637,065 confirmed cases of COVID-19 worldwide, including 5,148,221 deaths [5] (https://covid19.who.int/, accessed on 22 November 2021). As of 22 November 2021, a total of 7,370,902,499 vaccine doses had been administered worldwide [2]. Despite the tremendous milestones achieved in vaccine approval and administration, SARS-CoV-2, being an RNA virus, has genetically evolved over time, leading to the emergence of several variants from different geographic regions [6,7]. The variant strains have developed characteristics which grant them advantages in maintaining viral circulation, such as higher transmissibility and infectivity [8]. Most of these genetic differences are observed in the spike protein (S) region, specifically in the receptor-binding domain (RBD) and the N-terminal domain (NTD). The RBD and, to some extent, the NTD, as suggested by some evidence, are immunodominant, serving as the main neutralization targets for natural and vaccine-elicited antibodies [6,9]. The Delta variant was first reported in the Indian state of Maharashtra in December 2020 and harbors ten mutations (T19R, G142D, 156del, 157del, R158G, L452R, T478K, D614G, P681R, D950N) in the S protein [6]. Of note, the Delta variant lacks E484Q, a significant mutation associated with antibody neutralization resistance [10]. After successfully spreading globally, the prevalence of the Delta variant in the USA increased from 1.3% to 94.4% by 31 July 2021, while the Alpha variant decreased from 70% to 2.4% [7]. Perhaps of most serious concern, the Delta variant has been associated with breakthrough infections in vaccinated individuals [7]. The recent surge in cases despite extensive vaccination campaigns supports the concern about low vaccine effectiveness against variants. Studies are at odds regarding this topic, with some claiming that breakthrough infections are more likely to occur due to viral escape from antibodies [11,12]; others have demonstrated that mRNA vaccines remain effective through limiting COVID-19 severity, hospitalization and deaths [13,14,15,16,17,18,19,20,21]. Recently, more studies are exploring the efficacy of the natural immune response to SARS-CoV-2 vs. the mRNA vaccine-elicited response [22,23,24,25]. Our most recent work confirmed that following a natural infection, neutralizing antibody (nAbs) titers generated during infection accompanied by vaccination are significantly better in function than those generated by vaccination alone [26]. To this end, in this study, we compared the neutralization capacity of previously infected individuals and healthy previously unexposed individuals before and after vaccination against several variants of concern (VOCs) using a surrogate viral neutralization assay [27]. Our results from a Latino-origin population indicate that, compared with vaccination, natural infection induces a broader humoral response offering a wider range of protection against a rapidly evolving virus. These findings have pivotal implications in the understanding of the immune response to COVID-19 induced by vaccination amid emerging variants in the setting of a vaccinated population, and may contribute to future vaccine designs and booster schedules.

## 2. Materials and Methods

### 2.1. Study Samples

We selected individuals infected with SARS-CoV-2 at any time between March 2020 and February 2021. From 59 subjects followed for months, a subgroup of 10 vaccinated subjects previously exposed to SARS-CoV-2 and a subgroup of 21 healthy vaccinated volunteers who were never exposed to SARS-CoV-2 were followed for six to eight months. Vaccinated subjects received either the Pfizer-BioNTech or Moderna vaccine formulation. In the exposed group, all individuals tested positive for SARS-CoV-2 infection by quantitative PCR with reverse transcription (qRT–PCR) or serology tests (IgM and/or IgG). Serum samples from both groups were collected before vaccination (baseline), and after the first and second vaccine doses (Appendix A). Samples used in this study were obtained from adult volunteers (>21 years old) participating in the IRB- approved clinical protocol “Molecular basis and epidemiology of viral infections circulating in Puerto Rico”, Pro0004333. The protocol was submitted to, and ethical approval was given by, Advarra IRB on 21 April 2020. Participating volunteers were recruited before most of the SARS-CoV-2 variants were reported as circulating in Puerto Rico. More specifically, the Delta variant was first detected on 15 June 2021 [28].

### 2.2. cPass SARS-CoV-2 Neutralization Antibody Detection Assay

To determine the neutralizing activity of antibodies against SARS-CoV-2, we used a surrogate viral neutralization test (C-Pass GenScript sVNT, Piscataway, NJ, USA) according to the manufacturer’s instructions [13,26,27]. Briefly, serum samples were diluted according to the manufacturer’s instructions and incubated with soluble SARS-CoV-2 receptor-binding domain (RBD-HRP; S-RBD-HRP Wild Type (WT), Genscript Cat no. Z03594) antigen for 30 min at 37 °C, mimicking a neutralization reaction. Six different S-RBD-HRP-labeled mutants from Genscript (UK, B.1.1.7-Alpha, Cat no. Z03595; South Africa, B.1.351-Beta, Cat no. Z03596; Brazil P.1- Gamma, Cat no. Z03601; US, California, B.1.429/7-Epsilon, Cat no. Z03605; India B.1.617, Cat no. Z03608; India B.1.617.2 -Delta, Cat no. Z03614) were used in the assay, replacing the S-RBD-HRP WT component as variants of interest or concern to be assayed. Afterwards, samples were added to a 96-well plate coated with human ACE-2 protein. Following a 15 min incubation at 37 °C, RBD-HRP complexed with antibodies was removed in a wash step. The reaction was developed with tetramethylbenzidine (TMB) followed by a stop solution allowing the visualization of RBD-HRP bound to ACE2. Since this is an inhibition assay, color intensity was inversely proportional to the number of neutralizing antibodies present in the samples. Data were interpreted by calculating the percentage of inhibition of RBD-HRP binding. Samples with neutralization activity of ≥30% indicated the presence of SARS CoV-2 RBD-interacting antibodies capable of blocking the RBD–ACE2 interaction, thus inhibiting viral entry into host cells. While this assay measures the blocking activity of those antibodies, this activity is referred to throughout the text as “percentage of neutralization” for consistency and clarity.

### 2.3. Statistical Methods

Statistical analyses were performed using GraphPad Prism 7.0 software (GraphPad Software, San Diego, CA, USA). The statistical significance between or within groups was determined using two-way analysis of variance (ANOVA), one-way ANOVA (Tukey’s, Sidak’s or Dunnett’s multiple comparisons test as a post hoc test), an unpaired *t*-test or Wilcoxon–Mann–Whitney, to compare the means. The *p* values are expressed in relational terms with the alpha values. The significance threshold for all analyses was set at 0.05.

## 3. Results

### 3.1. Natural Infection Induces an Effective Neutralization against the Delta Variant

To examine the neutralization ability of sera from naturally infected individuals against the wild-type (WT) SARS-CoV-2, we evaluated baseline samples from 10 volunteers. Out of the 10 subjects, 8 had neutralizing activity greater than 70%, indicating the presence of antibodies capable of blocking the RBD-ACE2 binding (Figure 1A and Appendix A). The other two had neutralization degrees less than 70% but greater than 30%. To compare the neutralizing response elicited by WT SARS-CoV-2 to other virus strains, we exposed sera from those 10 individuals to six variants (Alpha, Beta, Gamma, Epsilon, Kappa and Delta). As expected, the highest neutralizing capacity observed was against the WT strain (Figure 1A). In comparison to the WT strain, there was a significantly decreased neutralizing activity against the Beta, Gamma and Kappa variants (*p* = 0.0041, *p* = 0.0003 and *p* = 0.0294, respectively). Surprisingly, no statistical differences were observed between the WT strain and the Alpha, Epsilon and Delta variants (Figure 1A). These results suggest that natural infection alone is capable of inducing a broad humoral response to various SARS-CoV-2 strains, including the Delta variant.

### 3.2. Vaccination Boosts Neutralizing Capacity against Variants in Previously Infected Individuals

To assess the humoral immune response to naturally acquired SARS-CoV-2 vs. the mRNA-based COVID-19 vaccine-elicited response, we compared the neutralizing capacity of exposed and unexposed subjects after one vaccine dose. Nineteen out of the twenty-one unexposed individuals (90.5%) produced nAbs (neutralization % >30) (Figure 1B and Appendix A). Similarly, all previously infected individuals reached neutralizing activity greater than 85% after just one vaccine dose (Figure 1C). This suggests that, in pre-exposed individuals, a single vaccine dose may be sufficient to grant protective immune status against WT SARS-CoV-2. When evaluating the neutralization from unexposed vaccinated individuals against the six VOCs, we found significant differences against all except the Delta variant, in comparison with the WT SARS-CoV-2 (*p* = 0.0075 for Alpha, *p* < 0.001 for Beta and Gamma, *p* = 0.0055 for Epsilon and *p* = 0.0012 for Kappa) (Figure 1B). This suggests that the Delta variant, in our population, does not escape neutralization by the antibodies induced by mRNA vaccination. In contrast, the neutralization activity in all previously exposed vaccinated individuals increased against all variants with no statistically significant differences (Figure 1C).

### 3.3. Full Vaccination Induces Limited Neutralizing Activity against All Tested Variants in Unexposed Individuals

Next, we evaluated the neutralizing capacity of antibodies after two vaccine doses in both previously exposed and unexposed individuals. All subjects (*n* = 31), regardless of immune status before vaccination, reached neutralization levels greater that 95% against WT SARS-CoV-2 after receiving a second vaccine dose (Figure 1D,E). This confirms that, in most COVID-19-naïve individuals, two vaccine doses are required to attain full protection. However, when exploring the neutralization against the variants, the unexposed individuals only gained similar neutralizing activity to the WT SARS-CoV-2 against the Epsilon and Delta variants (*p* = 0.0032 for Alpha, *p* < 0.001 for Beta and Gamma and *p* = 0.0035 for Kappa) (Figure 1D). Therefore, vaccination in unexposed individuals generates a neutralizing response against the Epsilon and Delta variants that is similar to the response against WT SARS-CoV-2, but only after the second dose. Highly relevant, even after the second dose, the neutralization against the other four variants was of a significantly lower magnitude compared to the WT.

On the other hand, we observed that the previously infected individuals maintained neutralizing capacity against all variants, similar to the response against the WT SARS-CoV-2 strain, denoting a key difference in the dynamics of vaccine-elicited antibodies between exposed and unexposed individuals (Figure 1E). This difference can be better appreciated in Figure 1F, where both vaccinated groups are compared after receiving the second dose. Of note, neutralization against the Alpha and Gamma variants did not behave similarly between groups, being of a higher magnitude in pre-exposed individuals (*p* = 0.0056 for Alpha and *p* < 0.0001 for Gamma) (Figure 1F).

## 4. Discussion

There is growing, but still limited, information available on immunity against the viral variants conferred by natural infection with the authentic SARS-CoV-2 strain or by the mRNA COVID-19 vaccines. Using samples collected during the COVID-19 pandemic, most of them before the documented introduction of the variants in the jurisdiction of Puerto Rico [26,28], we attempted to compare the kinetics of the nAbs response in the context of individuals with naturally acquired infection (pre-exposed) and unexposed individuals, following vaccination, via a widely used sVNT detecting RBD-targeting antibodies [26,29,30,31].

Strikingly, we found that natural infection before vaccination confers a broader neutralizing response against different SARS-CoV-2 strains, including the Delta variant, compared to the first dose of the COVID-19 mRNA vaccines. Using sera from individuals infected during wave 1 in the United Kingdom (UK), it was found that while the cross-neutralization of different SARS-CoV-2 VOC lineages is reduced, the sera neutralize the VOCs and parental virus to similar levels [24]. Another study found that a naturally infected individual produced uncommon genetic and structural characteristics and showed potent neutralization against authentic SARS-CoV-2 viruses, including VOCs [32]. These results are consistent with other reports [33,34,35] and highlight the need for more epidemiological data about the contribution of previously exposed individuals with naturally-acquired immunity to herd immunity. Overall, these issues are scarcely considered in any statistical model.

Highly relevant, our results also suggest that two vaccine doses may induce limited protection against some of the circulating variants in naïve individuals.

Consistent with other studies [34,36,37], our data confirm that subjects previously exposed to SARS-CoV-2 reach levels of protection against all tested variants after just one vaccine dose. Furthermore, we found a limited contribution, if any, of a second vaccine dose in pre-exposed individuals. Our results are also in line with a recent study characterizing the serum antibody classes and subclasses targeting the RBD of the S protein of SARS-CoV-2. They found that after primary vaccination, individuals with pre-existing immunity showed higher induction of all antibodies but IgG3 compared to SARS-CoV-2-naïve subjects, and that these antibodies targeted wild-type SARS-CoV-2 as well as its Alpha, Betta, Gamma and Delta variants. In agreement with our results, they also reported that pre-immune subjects did not mount a recall antibody response on receiving the second vaccine dose [25]. Together, our results and those from prior work reinforce the suggestion that individuals with a history of SARS-CoV-2 infection may not benefit from the second mRNA vaccine dose with the current standard regimen.

These findings strongly suggest that humoral immunity induced by natural infection results in higher-quality antibodies [34,35,37] and contributes to the expansion of memory B cells producing more cross-reactive antibodies following vaccination [35]. Andreano et al. dissected the nature of the memory B cell and antibody response at the single-cell level using samples from five naïve and five convalescent individuals vaccinated with the BNT162b2 mRNA vaccine. Consistent with our data, they found that the B.1.351 (Beta) and B.1.1.248 (Gamma) variants escaped almost seventy percent of the three thousand antibodies tested, in contrast to the B.1.1.7 (Alpha) and B.1.617.2 (Delta) variants, of which a much smaller portion were unaffected [22].

On the other hand, we found that in naïve subjects, a single dose of COVID-19 mRNA vaccines induced the same magnitude of nAbs against the Delta variant as against the WT strain. This response was improved after the second dose. However, even after a second dose, the magnitude of neutralization against other variants was significantly lower than against the WT strain.

A recent remarkable observational study in Puerto Rico collected hospitalization, death and vaccination rate data for more than 100,000 laboratory-confirmed SARS-CoV-2 infections over a period of 10 months. The study found that the effectiveness of the COVID-19 vaccines in preventing hospitalizations or death did not change after the Delta variant became dominant [28]. While that study did not segregate data at an individual level by the vaccination status of the SARS-CoV-2-positive individuals at the time of hospitalization or death, our results are perfectly aligned with, and provide the immunological rationale for, the findings of that study.

Recent studies suggested that the Delta variant may infect vaccinated individuals, creating what are defined as breakthrough infections [38]. In vitro neutralization results using monoclonal antibodies argue that vaccination induces a low level of nAbs against the Delta variant [11,35,39]. However, as demonstrated by Liu and colleagues, breakthrough infections by the Delta variant may be due to enhanced viral replication and infectivity, and not to antibody evasion or viral immune escape [7]. This statement is reinforced by the fact that the Delta variant lacks the E484Q mutation that seems to grant antibody resistance to other variants [9]. Thus, it seems that the Delta variant has developed the perfect evolutionary balance between transmissibility and virulence to become the dominant strain in circulation. However, there are limited or no data on breakthrough infections caused by the Delta variant in vaccinated people, comparing their prior immune status to SARS-CoV-2.

Our findings, together with prior reports on the effectiveness of the cellular immune response against the variants [35,40,41,42,43], warrant a revision of COVID-19 vaccine policy implementation in subjects with prior natural immunity to SARS-CoV-2.

Since early during the pandemic, patients of older age (>65 years old) and with comorbidities, such as diabetes, respiratory disease and coronary heart disease, have had the worst prognosis, as these factors seem to have an impact on COVID-19 disease outcomes [44,45]. Additionally, these individuals may have weaker immune systems by default that could give rise to breakthrough infections, but vaccination could still contribute to a lower mortality risk. Despite the progress achieved with the introduction of COVID-19 vaccines, some hesitancy has emerged among the population during the last year regarding their safety and efficacy. Vaccine acceptance has lessened due to social and human factors, such as misinformation in social media and lack of public health impact in communities [46]. Nonetheless, vaccination is still considered the most effective way of preventing severe disease and mortality. Vaccination, along with natural herd immunity, is our ticket to restoring our lives to some sense of normalcy, and it needs to advance, along with epidemiological and genomic surveillance, as a means to counter progressing SARS-CoV-2 fitness [38].

We are aware of the limitations of our study, including the assay we implemented. The presence of non-RBD antibodies possessing neutralizing capacity has been documented [47,48]. However, as documented in the literature, the RBD domain continues to be the key target for SARS-CoV-2 neutralization [49,50,51,52,53]. The relevance of the RDB domain as a main target of neutralization is also supported by a nanoparticle-based vaccine platform for the multivalent display of the RBD. This vaccine formulation induced broadly cross-reactive antibodies with high neutralizing properties not only against an early isolate of SARS-CoV-2 but also against three SARS-CoV-2 variants of concern, including the Delta variant, as well as SARS-CoV-1 [54]. We cannot rule out an exceptional contribution of some unique non-RBD nAbs. Those antibodies may not be detected by the surrogate assay we implemented but taking into account prior work using the same sVNT assay, we believe we are capturing the big picture in terms of neutralization. Another limitation is the small sample size and lack of cellular immunity characterization. The waning of natural or vaccine-elicited immunity remains a possibility outside the follow-up period used in this study. However, our results and those from others [21,25], despite being obtained from a population of different genetic backgrounds, agree with the current ongoing scenario (October 2021) in the UK where a high level of vaccine effectiveness against symptomatic diseases with the Delta variant was found among individuals who had received two COVID-19 vaccines doses [55]. In that country, a rampant increase in Delta variant circulation, up 35% over the two previous weeks, was observed after all restrictions were lifted in summer 2021 [56]. In spite of this, taking into account the high number of cases naturally exposed to the virus and the high vaccination rate in the UK [57], as would be anticipated from our results, the daily deaths were a tenth of what they were in the prior wave [56,58]. Considering our findings and those from other groups [13,14,15,18,21], a more challenging scenario would be a predominance of other variants such as Alpha, Beta, Gamma or Kappa, showing limited neutralization after full vaccination with the mRNA COVID-19 vaccines. These results warrant further follow-up by public health institutions and officials to develop a preparedness plan in anticipation of the predominance of less effectively controlled variants. After the largest viral pandemic thus far of modern times, the “Spanish flu” (estimated total deaths from 50 to 100 million) [59,60,61,62], major genetic changes were needed in the influenza virus to cause new outbreaks [63,64]. However, none of them were of the magnitude in terms of morbidity or mortality of the 1918 pandemic. With that in mind, it is reasonable to suggest that as of November 2021, two years after the detection of SARS-CoV-2 for the first time, the immune system at a population level is in a better position to recognize and fight back more effectively against any newly arising variant of this virus.

To our knowledge, this is the first study conducted in a Hispanic/Latino population impacted by COVID-19. Our findings are a significant contribution to the still lacking population-based studies concerning virus population dynamics in the setting of vaccination and shed light on the design of second-generation COVID-19 vaccines.

## Figures and Tables

**Figure 1 viruses-13-02405-f001:**
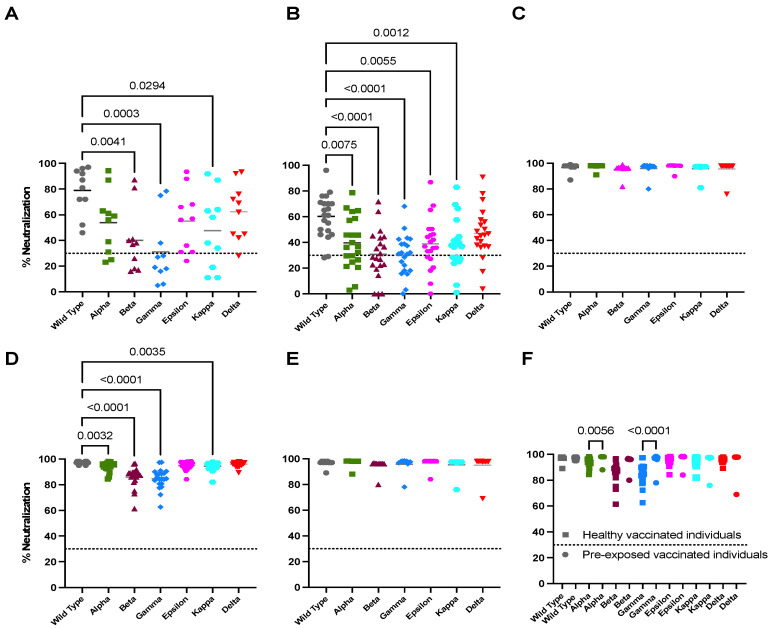
Neutralization capacity of sera from infected and non-infected individuals against SARS-CoV-2 variants before and after vaccination. The neutralization activity of sera from infected individuals (*n* = 10) and non-infected ones (*n* = 21) before and after vaccination was evaluated against the six variants of concern. Dotted lines indicate the limit of detection of the sVNT assay, where the percentage of signal inhibition is determined (≥30% indicates a positive result). A normality test (Shapiro–Wilk) was performed for all datasets in order to assess the distribution of the data. The significance threshold for all analyses was set at *p* < 0.05. (**A**). Neutralization activity of sera from infected individuals (*n* = 10) before vaccination. A one-way ANOVA test with Dunnett’s multiple comparisons test was performed between each of the variants. (**B**). Neutralization activity of sera from healthy individuals (*n* = 21) after receiving the 1st vaccine dose. A one-way ANOVA test with Dunn’s Kruskal–Wallis multiple comparisons test was performed between each of the variants. (**C**). Neutralization activity of sera from infected individuals (*n* = 10) after receiving the first vaccine dose. A one-way ANOVA test with Dunnett’s multiple comparisons test was performed between each of the variants. (**D**). Neutralization activity of sera from healthy individuals (*n* = 21) after receiving the 2nd vaccine dose. A one-way ANOVA test with Dunn’s Kruskal–Wallis multiple comparisons test was performed between each of the variants. (**E**). Neutralization activity of sera from infected individuals (*n* = 10) after receiving the 2nd vaccine dose. A one-way ANOVA test with Dunnett’s multiple comparisons test was performed between each of the variants. (**F**). Neutralization activity of sera from vaccinated individuals, pre-exposed (*n* = 10, depicted in circles) and healthy (*n* = 21, depicted in squares), after receiving the 2nd dose was evaluated. A one-way ANOVA test with Dunn’s Kruskal–Wallis multiple comparisons test was performed between each of the variants.

## Data Availability

All data are available upon request.

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
