# Peer review of "Limited Impact of Delta Variant’s Mutations on the Effectiveness of Neutralization Conferred by Natural Infection or COVID-19 Vaccines in a Latino Population"

_viruses, 2021, doi:10.3390/v13122405_

Round 1
Reviewer 1 Report
The manuscript by Sariol et al. describes an analysis of serum neutralizing responses to SARS-CoV-2 mRNA vaccination by individuals who were either healthy or became infected with COVID-19 prior to vaccination. The study shows that individuals with prior SARS-CoV-2 exposure produced a more potent and consistent neutralizing response compared to previously non-exposed counterparts. The study appears well designed and, importantly, included key controls comparing titers both before and after vaccination.
Minor points:
The authors suggest a few times throughout the manuscript that there is uncertainty in the literature surrounding the effects of prior SARS-CoV-2 exposure on vaccination. On the contrary, I would say there have recently been a number of high-profile studies that have suggested very similar to data to those presented in this manuscript, and therefore these sections should be corrected to reflect this.
Notably the authors used a neutralization assay kit that only measures binding to the SARS-CoV-2 spike RBD. Numerous works have revealed neutralizing responses targeted to viral epitopes in close proximity yet external to RBD. Do the authors believe this may have had an effect upon results presented here? I believe some comment or discussion in this area would be strongly beneficial.
Author Response
Reviewer 1
The manuscript by Sariol et al. describes an analysis of serum neutralizing responses to SARS-CoV-2 mRNA vaccination by individuals who were either healthy or became infected with COVID-19 prior to vaccination. The study shows that individuals with prior SARS-CoV-2 exposure produced a more potent and consistent neutralizing response compared to previously non-exposed counterparts. The study appears well designed and, importantly, included key controls comparing titers both before and after vaccination.
Authors thanks the reviewer for the positive feedback.
Minor points:
The authors suggest a few times throughout the manuscript that there is uncertainty in the literature surrounding the effects of prior SARS-CoV-2 exposure on vaccination. On the contrary, I would say there have recently been a number of high-profile studies that have suggested very similar to data to those presented in this manuscript, and therefore these sections should be corrected to reflect this.
We apologize for the oversight. Initially we wanted to provide this data as a short report and limited number of references were cited. We reviewed the topic and now we added new information. We also change the text to reflect the reviewer’s suggestion and available data. This was modified in the introduction (lines 52-57) and in the discussion as well.
In the initial sentence of the discussion (lines 200-202) we changed the wording:
From:
There is still limited information available on the immunity conferred by the natural infection with the authentic SARS-CoV-2 strain or the mRNA COVID-19 vaccines against the viral variants.
to
There is growing, but still limited information available on the immunity conferred by the natural infection with the authentic SARS-CoV-2 strain or the mRNA COVID-19 vaccines against the viral variants.
Notably the authors used a neutralization assay kit that only measures binding to the SARS-CoV-2 spike RBD. Numerous works have revealed neutralizing responses targeted to viral epitopes in close proximity yet external to RBD. Do the authors believe this may have had an effect upon results presented here? I believe some comment or discussion in this area would be strongly beneficial.
We thank the reviewer for picking this up. We provided a more extensive comment on this in our recent publication in Viruses https://www.mdpi.com/1999-4915/13/10/1972. This is why we only referenced that on this manuscript.
However in the reviewed version we added additional supporting information on the dominance of the RBD as a major target for neutralization.
Reviewer 2 Report
Authors wrote an very interesting paper, well wrote and well presented.
Only some minor suggestions
- Introduction: updata data on SARS CoV2 wordwilde. Furthermore, add the indirect effect of SARS CoV2 on other diseases with increase in diagnostic delay and incresase clinical severity (see and cite Increase in Tuberculosis Diagnostic Delay during First Wave of the COVID-19 Pandemic: Data from an Italian Infectious Disease Referral Hospital. Antibiotics (Basel). 2021 Mar 8;10(3):272. doi: 10.3390/antibiotics10030272.)
- Methods and results: well presented
- Discussion: discuss better the role of vaccination, and also the role of acceptance of vaccination and the hesitancy also in global health approach (see Vaccination among Healthcare Workers: Results from a National Survey in Italy. Viruses. 2021 Feb 26;13(3):371. doi: 10.3390/v13030371. ). In addiction, discuss also the role of comorbidity and age on covid impact and mortality.
- COnclusion : give some global health proposal that came from your interesting and very well wrote paper
Author Response
Authors wrote an very interesting paper, well wrote and well presented.
Authors thank the reviewer for the positive feedback on our work.
Only some minor suggestions
- Introduction: updata data on SARS CoV2 wordwilde. Furthermore, add the indirect effect of SARS CoV2 on other diseases with increase in diagnostic delay and incresase clinical severity (see and cite Increase in Tuberculosis Diagnostic Delay during First Wave of the COVID-19 Pandemic: Data from an Italian Infectious Disease Referral Hospital. Antibiotics (Basel). 2021 Mar 8;10(3):272. doi: 10.3390/antibiotics10030272.)
Data was updated accordingly, and manuscript was cited.
- Methods and results: well presented
Authors thank you the reviewers for the encouraging comment.
- Discussion: discuss better the role of vaccination, and also the role of acceptance of vaccination and the hesitancy also in global health approach (see Vaccination among Healthcare Workers: Results from a National Survey in Italy. Viruses. 2021 Feb 26;13(3):371. doi: 10.3390/v13030371. ). In addiction, discuss also the role of comorbidity and age on covid impact and mortality.
Data was updated accordingly, and manuscript was cited.
- COnclusion : give some global health proposal that came from your interesting and very well wrote paper
We provided a conclusion related to the real time situation going on in UK at the time we wrote the manuscript. Following the reviewer’s suggestion, we wanted to provide a parallel of the variant’s emergence and the evolution of the flu virus since the biggest documented viral pandemic in 1918.